# A Novel Artificial Intelligence Technique to Estimate the Gross Calorific Value of Coal Based on Meta-Heuristic and Support Vector Regression Algorithms

**Hoang-Bac Bui [1,2], Hoang Nguyen [3,*], Yosoon Choi [4,*], Xuan-Nam Bui [5,6], Trung Nguyen-Thoi [7,8] and Yousef Zandi [9]**

1. Faculty of Geosciences and Geoengineering, Hanoi University of Mining and Geology, 18 Vien street, Duc Thang ward, Bac Tu Liem District, Hanoi 100000, Vietnam; buihoangbac@humg.edu.vn
2. Center for Excellence in Analysis and Experiment, Hanoi University of Mining and Geology, 18 Vien street, Duc Thang ward, Bac Tu Liem District, Hanoi 100000, Vietnam
3. Institute of Research and Development, Duy Tan University, Da Nang 550000, Vietnam
4. Department of Energy Resources Engineering, Pukyong National University, Busan 48513, Korea
5. Department of Surface Mining, Mining Faculty, Hanoi University of Mining and Geology, 18 Vien Street, Duc Thang Ward, Bac Tu Liem District, Hanoi 100000, Vietnam; buixuannam@humg.edu.vn
6. Center for Mining, Electro-Mechanical Research, Hanoi University of Mining and Geology, 18 Vien Street, Duc Thang Ward, Bac Tu Liem District, Hanoi 100000, Vietnam
7. Division of Computational Mathematics and Engineering, Institute for Computational Science, Ton Duc Thang University, Ho Chi Minh City 70000, Vietnam; nguyenthoitrung@tdtu.edu.vn
8. Faculty of Civil Engineering, Ton Duc Thang University, Ho Chi Minh City 70000, Vietnam
9. Department of Civil Engineering, Tabriz Branch, Islamic Azad University, Tabriz 51368, Iran; zandi@iaut.ac.ir
* Correspondence: nguyenhoang23@duytan.edu.vn (H.N.); energy@pknu.ac.kr (Y.C.)

**Abstract:** Gross calorific value (GCV) is one of the essential parameters for evaluating coal quality. Therefore, accurate GCV prediction is one of the primary ways to improve heating value as well as coal production. A novel evolutionary-based predictive system was proposed in this study for predicting GCV with high accuracy, namely the particle swarm optimization (PSO)-support vector regression (SVR) model. It was developed based on the SVR and PSO algorithms. Three different kernel functions were employed to establish the PSO-SVR models, including radial basis function, linear, and polynomial functions. Besides, three benchmark machine learning models including classification and regression trees (CART), multiple linear regression (MLR), and principle component analysis (PCA) were also developed to estimate GCV and then compared with the proposed PSO-SVR model; 2583 coal samples were used to analyze the proximate components and GCV for this study. Then, they were used to develop the mentioned models as well as check their performance in experimental results. Root-mean-squared error (RMSE), correlation coefficient ($R^2$), ranking, and intensity color criteria were used and computed to evaluate the GCV predictive models developed. The results revealed that the proposed PSO-SVR model with radial basis function had better accuracy than the other models. The PSO algorithm was optimized in the SVR model with high efficiency. These should be used as a supporting tool in practical engineering to determine the heating value of coal seams in complex geological conditions.

**Keywords:** gross calorific value; coal; proximate analyze; artificial intelligence; PSO-SVR

## 1. Introduction

Coal is one of the non-renewable natural resources, like oil and gas [1]. It is a fossil fuel that is widely used in the metallurgical industry and in thermal power plants [2,3]. Besides, it is also used for cement industries, industrial chemicals, and aluminum, to name a few [4]. So far, coal still accounts for 40% to 50% of energy generation in the world [5]. The main components of coal include carbon (C), hydrogen (H), nitrogen (N), sulfur (S) and oxygen (O). For the chemical parameters, fixed carbon (FC), ash (A), volatile matter (VM), and moisture (M) are the necessary components of coal [6]. These are the main parameters that decide the heat value of coal [7,8]. A literature review indicated that most of coal applications are related to heating value (gross calorific value-GCV) [9]. Depending on each specific field (e.g., steel industry, cement, power plant, to name a few), the GCV requirements are different [10]. Therefore, determination of GCV in coal seams essential for calculating energy demand, and improving economic effectiveness, as well as selective extraction.

In recent years, scientific and technological issues related to energy and fuel have achieved many new achievements. Simulation and optimization methods for the use of fuel have also been proposed [11–20]. Environmental issues due to fuel impacts have also been considered and studied [21–25]. Among the types of fuel, the GCV of coal is considered as a primary fuel type for thermal plans. For determination coal GCV, many scholars have been carried out to investigate the relationship between GCV and proximate analysis (i.e., FC, A, VM, and M) as well as coal properties [26–28]. For instance, many equations and soft computing models have been proposed. Empirical methods for estimating GCV of coal have been introduced very early [29–32]. However, they usually take a long time with low accuracy. In recent years, artificial intelligence (AI) is well-known as an advanced technique to solve most of the complex problems relevant engineering, especially in the field of energy and fuels [33–51]. For predicting the GCV of coal, Akkaya [52] applied multiple nonlinear regression models for predicting GCV of coal with high reliability. Tan et al. [28] also developed a soft computing model for predicting GCV based on support vector regression (SVR) algorithm with a promising result. In another study, an artificial neural network (ANN) successfully studied and applied in predicting GCV by Mesroghli et al. [53]. They concluded that the ANN model was suitable for estimating GCV with high accuracy. Erik Yilmaz [54] also developed ANN and an adaptive neuro–fuzzy inference mode (ANFIS) to simulate the GCV of coal. In their study, the ANFIS model exhibited accuracy better than the ANN model. Another technique using wavelet neural networks (WNNs) was also introduced and was applied to estimate GCV by Wen et al. [55]. They compared their WNNs model with previous methods (employed and redeveloped previous models) to develop a complete conclusion. Their results indicated that the WNNs model yielded higher accuracy than the earlier methods in their study. By a new approach, Wood [56] developed a transparent open-box learning networks (TOB) algorithm for predicting the GCV of coal. He found that the TOB algorithm can predict GCV with a tiny error. In addition, many similar works were also carried out to predict GCV using AI techniques, such as [57–65].

As in our review of GCV prediction, many soft computing models were widely developed and applied during recent years. However, their performance was not confirmed in other areas since the coal quality in each region/area/seam was different. Furthermore, new soft computing/AI models, with the capability of accurate prediction of GCV, have become the goal of scholars. Therefore, this study aims to reach several novelties with scientific soundness, and can be summarized as follow:

- Big data with 2583 coal samples were used to analyze proximate components and the GCV for this study.
- A meta-heuristic algorithm (i.e., a particle swarm optimization (PSO) algorithm) was considered to optimize the support vector regression (SVR) models for predicting GCV. Subsequently, the particle swarm optimization (PSO)-support vector regression (SVR) model with the radial basis kernel function was proposed as the best model for the aiming of GCV prediction.

-     A variety of different AI models were also developed to predict GCV and compared with the proposed PSO-SVR model, including classification and regression trees (CART), multiple linear regression, and principles component analysis (PCA).

## 2. Study Area and Data Properties

This study was undertaken in an underground coal mine in Quang Ninh province, Vietnam. It lies geographically between longitudes 106°36'23" E and 106°36'47" E; latitude 21°04'10" N and 21°04'30" N (Figure 1). The study site shows a complex geological structure with many faults and folds. The results of stratigraphic studies indicated that the mine area consists of coal-bearing sediments of Hon Gai Formation (T3n-r hg) with a thickness of 600–800 m (Figure 1). The coal seams have different thickness alternating with layers of gravel, sandstone, sandstone, siltstone, and claystone. Coal in the study site belongs to anthracite to bituminous coal. It is black and has black and gray markings, glass or metallic luster. The coal is mainly homogeneous, brittle, and less cracked. The previous results indicated that the study site coal has a high calorific value, low volatile matter, and sulfur content, with an excellent response to industries.

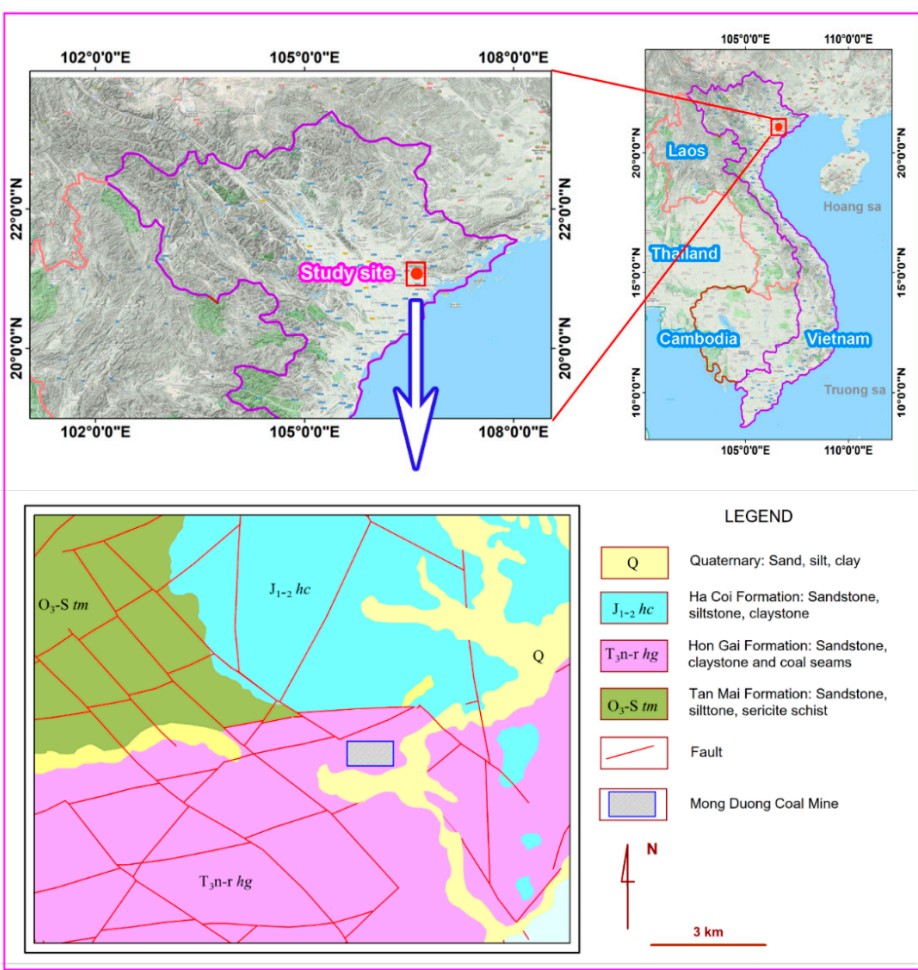

**Figure 1.** Location and geological conditions of the study area.

In this study, the database was collected from previous geological reports at the Mong Duong coal mine for many years [66]. The database consisted of the results of the proximate and ultimate analysis along with the gross calorific values from the coal samples taken at different coal seams in the mine. Sampling, processing, and analyzing methods were carried out by Vietnam's standards (TCVN) [67–70]. In this study, the results of the proximate analysis for coal samples were used to predict the GCV. The values of the moisture content (M), ash (A), and volatile matter (VM) for each sample

were measured in the laboratory. According to Patel et al. [71], fixed carbon (FC) was considered as an input variable for predicting GCV. However, FC can be calculated by the given equation of FC = 100 − (M + A + VM). Therefore, the correlation between FC and M, A, and VM is 100%. As per recommended in statistical techniques, high correlation parameters should be removed from the dataset to avoid effect on the accuracy of the models [72,73]. Matin, Chelgani [74] were also given similar recommendations and the FC was also not used in their study for predicting GCV. Therefore, FC was not used to predict GCV in this study. Herein, a dataset of 2583 coal samples used in this study included factors of M, A, VM, and GCV. A brief of the dataset used in this study is listed in Table 1. In addition, an illustration of the dataset used in this study is shown in Figure 2.

**Table 1.** Brief of the gross calorific value (GCV) data used.

| Category | Moisture (M, %) | Ash (A, %) | Volatile Matter (VM, %) | GCV (Kcal/kg) |
|---|---|---|---|---|
| Min. | 0.200 | 1.32 | 3.580 | 4352 |
| 1st Quarter | 1.520 | 8.95 | 6.435 | 6128 |
| Median | 2.000 | 17.70 | 7.800 | 6816 |
| Mean | 2.037 | 17.60 | 7.860 | 6825 |
| 3rd Quarter | 2.540 | 24.89 | 9.175 | 7625 |
| Max. | 4.350 | 39.96 | 11.990 | 8654 |

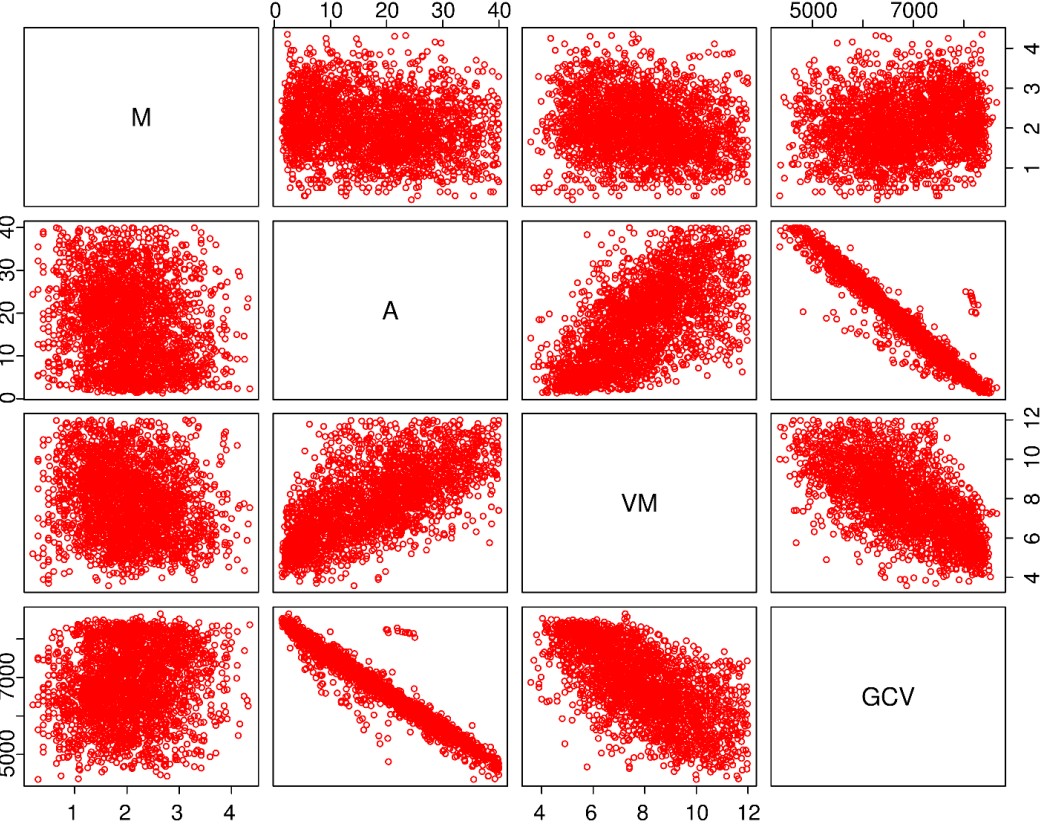

**Figure 2.** Visualization of the data used in this study. M—moisture content; A—ash; VM—volatile matter; GCV—gross calorific value.

## 3. Methods

### 3.1. Support Vector Regression (SVR)

The name SVR indicates a well-known notion of one of the most commonly used AI techniques, which was first proposed by Vapnik [75]. As well as SVR, SVC (support vector classification) is another

type of SVM (support vector machine). However, SVR is known as the most commonly held tool for simulating continuous problems. More details about the SVR are available in previous studies such as [76–79]. Moreover, three kernel functions which are suitable for regression problems are "linear function", "polynomial function", and "radial basis function" that are formulated in Equations (1)–(3), respectively:

$$F(X_{GCV}, Y_{GCV}) = X_{GCV}^T Y_{GCV} \tag{1}$$

$$F(X_{GCV}, Y_{GCV}) = (\gamma \cdot X_{GCV}^T Y_{GCV} + r)^d \; ; \gamma > 0; \; d = (1, 2, \dots, n) \tag{2}$$

$$F(X_{GCV}, Y_{GCV}) = \exp[\frac{\|X_{GCV} - Y_{GCV}\|^2}{2\sigma^2}] \tag{3}$$

where *n* is the number of observations; *X* and *Y* denote the input and output variables, respectively; *r*, *d*, $\gamma$, and $\sigma$ denote the hyper-parameters of the SVR models. In addition, *C* (cost) is considered as a penalty parameter which is also used to control the quality of the SVR models.

### 3.2. Particle Swarm Optimization (PSO)

The PSO algorithm is known as one of the powerful optimization algorithms, which was introduced by Eberhart, Kennedy [80]. Inspired by the behavior of the particles/social animals, PSO simulates the movement of a fish flock or choreography of a flock bird, or to a swarm of insects. It is explained based on the feeding instincts of animals. They tend to follow individuals that lead to optimal sources of food [81]. In addition, the process of exchanging information between individuals is carried out continuously during the migration process to point to the most optimal source of food. In addition, more details in the PSO algorithm can be found in the following literature [82–86]. The pseudo-code of the PSO algorithm is described as follows [87]:

| **Algorithm:** The pseudo-code of PSO (particle swarm optimization) algorithm |
|---|

| | |
|---|---|
| 1 | **for** each particle *i* |
| 2 |    **for** each dimention *d* |
| 3 |       Initialize position $x_{id}$ randomly within permissible range |
| 4 |       Initialize velocity $v_{id}$ randomly within permissible range |
| 5 |    **end for** |
| 6 | **end for** |
| 7 | Iteration *k* = 1 |
| 8 | **do** |
| 9 |    **for** each particle *i* |
| 10 |       Calculate fitness value |
| 11 |       **if** the fitness value is better than $p\_best_{id}$ in history |
| 12 |          Set current fitness value as the $p\_best_{id}$ |
| 13 |       **end if** |
| 14 |    **end for** |
| 15 |    Choose the particle having the best fitness value as the $g\_best_{id}$ |
| 16 |    **for** each particle *i* |
| 17 |       **for** each dimention *d* |
| |          Calculate velocity according to the following equation |
| 18 | $v_j^{i+1} = wv_j^{(i)} + (c_1 \times r_1 \times (local\ best_j - x_j^{(i)})) + (c_2 \times r_2 \times (global\ best_j - x_j^{(i)})), v_{\min} \le v_j^{(i)} \le v_{\max}$ |
| |          Update particle position according to the following equation |
| 19 | $x_j^{i+1} = x_j^{(i)} + v_j^{(i+1)}; \; j = 1, 2, \dots, n$ |
| 20 |       **end for** |
| 21 |    **end for** |
| 22 | *k* = *k*+1 |
| 23 | **while** maximum iterations or minimum error criteria are not attained |

### 3.3. PSO-SVR Model for Estimating GCV

As the primary goal of this study, the new hybrid artificial intelligence model PSO-SVR is described and offered in this section. It is a combination of the PSO and SVR algorithms for the generation of an optimal model in predicting GCV. Accordingly, SVR was determined as the primary model for predicting GCV. For the development of the SVR model, three kernel functions (i.e., radius basis function (RBF), Polynomial (P), and Linear (L)) were applied to map the GCV database. For each kernel function, hyper-parameter(s) were determined, as listed in Table 2. The primary purpose of the hyper-parameters was controlling the accuracy of the SVR model. Therefore, the selection of the optimal values of hyper-parameters was a complication, and it needed a solution for defining the optimal hyperparameters. In this case, the PSO algorithm was an optimal solution for searching optimal hyper-parameters of the SVR model. It performed a global search solution based on particles and their experiences. For each value of hyper-parameter searched, it calculated the performance of the SVR model through a fitness function (i.e., root-mean-squared error (RMSE)). Then, it checked the termination criteria. If the value of RMSE was satisfied (lowest RMSE), it stopped the searching process and gave the final PSO-SVR model. Otherwise, it continued searches until it was satisfied, or it reached the maximum of iterations. Figure 3 describes the framework of the proposed PSO-SVR model with three forms of kernel functions.

**Table 2.** The support vector regression (SVR) hyper-parameters with various kernel functions.

| PSO-SVR Model | Hyper-Parameters | | | |
|---|---|---|---|---|
| | $C$ | $d$ | $\gamma$ | $\sigma$ |
| Linear function | x | - | - | - |
| Polynomial function | x | x | x | - |
| Radial basis function | x | - | - | x |

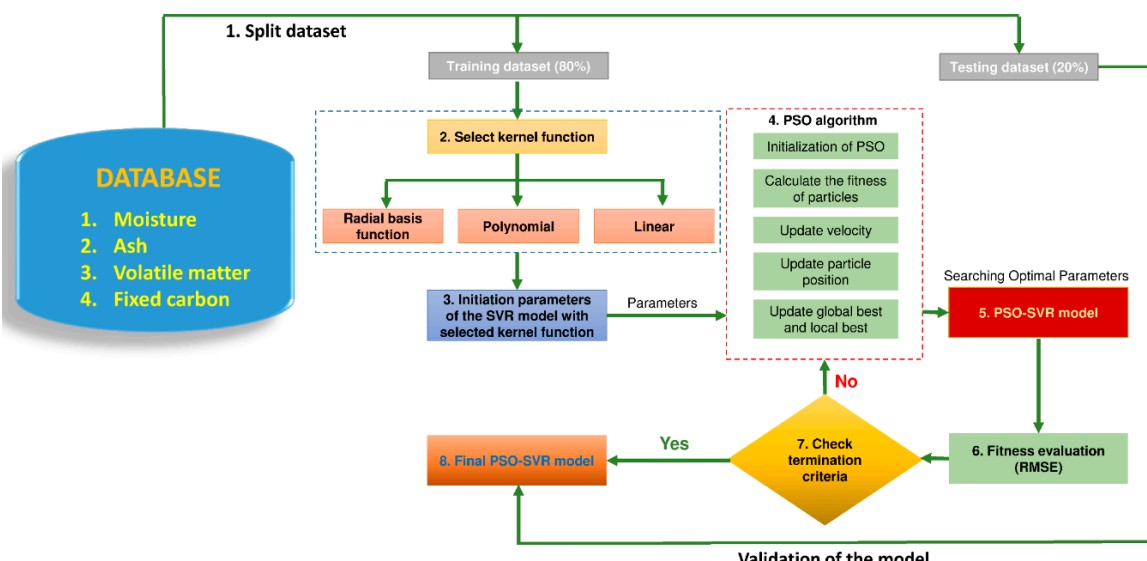

**Figure 3.** The proposed PSO-SVR framework for predicting GCV in this study.

### 3.4. Multiple Linear Regression (MLR)

MLR is one of the simply-implemented regression models used to predict direct correlations between two or more variables. It works on the principle of analyzing the relationship between the independent (i.e., influential) variables and dependent (i.e., target) variables [88]. The general equation of MLR is described as follows:

$$Y = a_1 X_1 + a_2 X_2 + \ldots + a_n X_n + b \tag{4}$$

where $X_1$ to $X_n$ represents the independent factors; *n* symbolizes the number of independent factors; *Y* is the dependent variable; $a_1$ to an are the correlation coefficients of independent variables and dependent variables; *b* is the deviation value.

### 3.5. Classification and Regression Tree (CART)

CART is a famous notion of decision tree approaches in data mining, which was proposed by Breiman [89] (this book was introduced first in 1984, and reprinted in 2017). The two main advantages of the CART algorithm include:

- Ability to explain the rules created quickly;
- Ability to apply for both classification and regression problems.

For the regression problem, CART performs separation rules in the form of "$X \leq C$?" or "$X \geq C$?" [90]. Its separation mechanism is primarily based on the parent's node to separate duality into two children nodes [91]. The separation process can be conducted in multiple branches until the initial condition is satisfied. The pseudo-code of the CART algorithm can be found in [92].

### 3.6. Principles Component Analysis (PCA)

PCA is one of the multivariate data processor techniques, which was suggested by Wold et al. [93]. The nature of PCA is to implement a dimensionality reduction procedure. This method is based on the observation that data is often not distributed randomly in space but is often distributed near specific lines/faces [94]. PCA considers a particular case when those different faces are linear forms of subspaces [95]. For PCA modeling, seven steps are implemented, as illustrated in Figure 4.

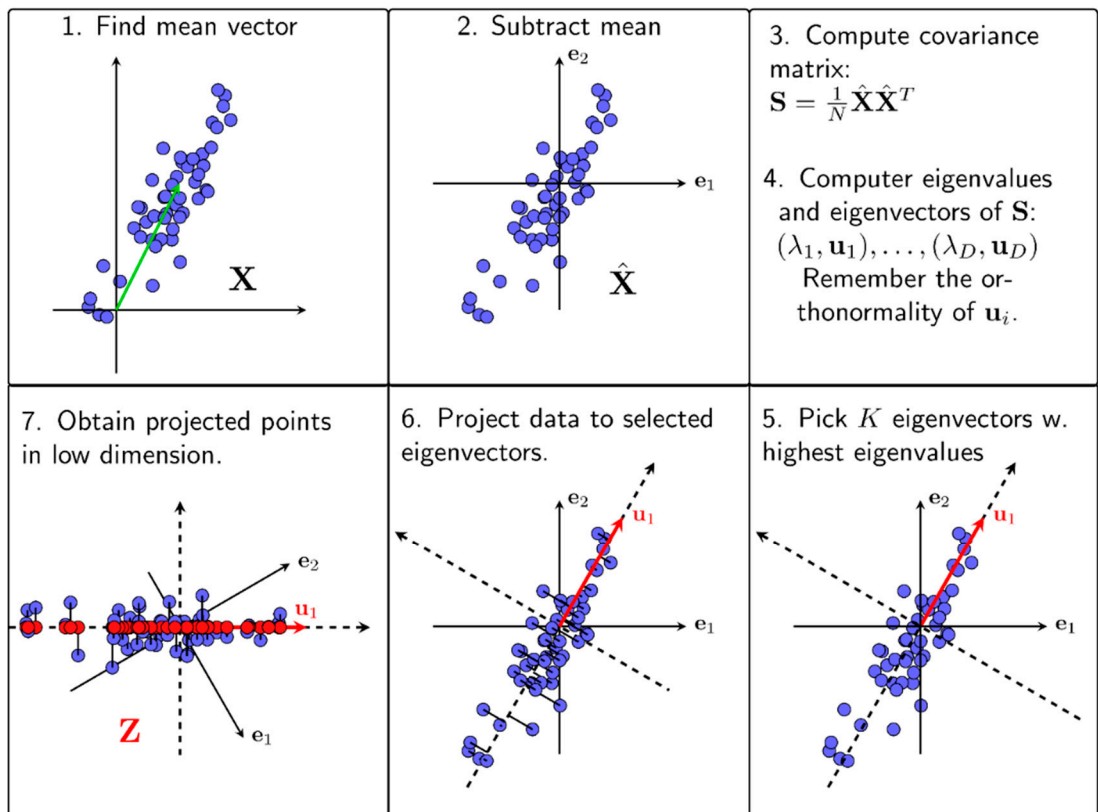

**Figure 4.** The principle component analysis (PCA) procedure (source: https://machinelearningcoban. com/2017/06/15/pca/).

## 4. Model Assessment Indices

Before developing the models, assessment indices were needed to evaluate their accuracy, as well as the error level of the model. Herein, two benchmark statistical criteria were used, including $R^2$ and *RMSE*. They were computed as:

$$RMSE = \sqrt{\frac{1}{n}\sum_{k=1}^{n}\left(y_{GCV} - \hat{y}_{GCV}\right)^2}$$ (5)

$$R^2 = 1 - \frac{\sum_k \left(y_{GCV} - \hat{y}_{GCV}\right)^2}{\sum_k \left(y_{GCV} - \overline{y}\right)^2}$$ (6)

where $n$ is the number of samples; $y_k$, $\hat{y}_k$ and $\overline{y}$ are measured, predicted, and the mean of $y_k$ values, respectively.

## 5. Results and Discussion

Before developing the mentioned models, the dataset needed to prepare and normalize. According to previous research, the dataset should be divided into two groups with an 80/20 ratio [96–98]. Hence, a split procedure with 80% of the whole dataset (~2327 samples) was used for training. Then, the remaining 20% (~256 samples) was used for evaluating the developed predictive models. Note that all mentioned models were generated based on the same training dataset, and tested based on the same testing dataset. RMSE and $R^2$ were used to evaluate their performance on both training and testing datasets.

### 5.1. PSO-SVR Models

As stated above, the proposed PSO-SVR model was tried with the three kernel functions (i.e., RBF, P, and L) and their parameters, as shown in Table 2. For implementing the PSO-SVR models, the framework in Figure 6 was applied. The PSO algorithm and its parameters were needed to set up before searching the optimal hyperparameters of the SVR models. In which, the swarm size (s), maximum particle's velocity ($V_{max}$), the maximum number of iterations (i), individual and group cognitive parameters ($\phi_1$, $\phi_2$), and inertia weight ($w$) were the parameters of PSO, which were used for the optimization process. According to previous works, s should be enough to ensure diversity [35,99,100]; thus, the swarm size of 100, 200, 300, 400, 500, respectively, was applied for the PSO procedure. The other parameters (i.e., Vmax, $\phi_1$, $\phi_2$, and $w$) was set to 2.0, 1.8, 1.8, and 0.9, respectively, according to recommendations of previous researchers [101–104]. To check the termination criteria of the optimization process as well as ensure the optimal search process, $i$ was set to 1000 iterations.

Once the parameters of the PSO algorithm were established, the SVR models with various kernel functions were generated with initiation parameters. Then, the PSO procedure performed searching optimal values according to the hyper-parameters in Table 2. Finally, the optimal values of the hyper-parameters were determined with the lowest RMSE, and the PSO global search procedure was stopped after 1000 iterations. Table 3 shows the optimal values of the hyper-parameters with respectively kernel functions. In addition, Figures 5–7 show the performance of the optimization process based on kernel functions (i.e., PSO-SVR-L; PSO-SVR-P; PSO-SVR-RBF).

**Table 3.** The optimal values of the PSO-SVR hyper-parameters.

| Model | Hyper-Parameters | | | |
|---|---|---|---|---|
| | $C$ | $d$ | $\gamma$ | $\sigma$ |
| PSO-SVR-L | 147.025 | - | - | - |
| PSO-SVR-P | 0.157 | 3 | 0.870 | - |
| PSO-SVR-RBF | 4.567 | - | - | 0.279 |

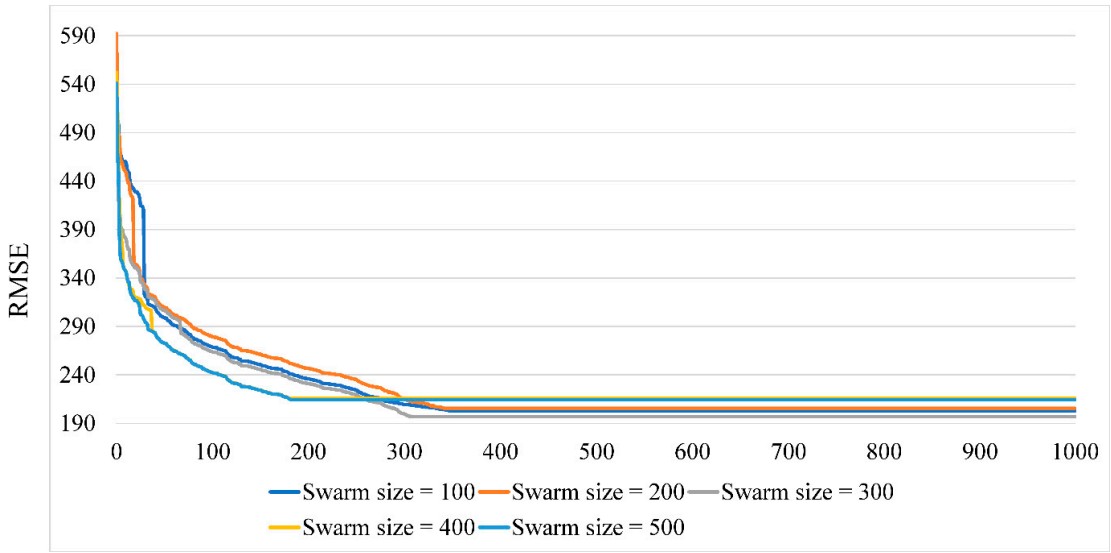

**Figure 5.** Performance of the PSO-SVR ensemble with the RBF kernel function on the training process.

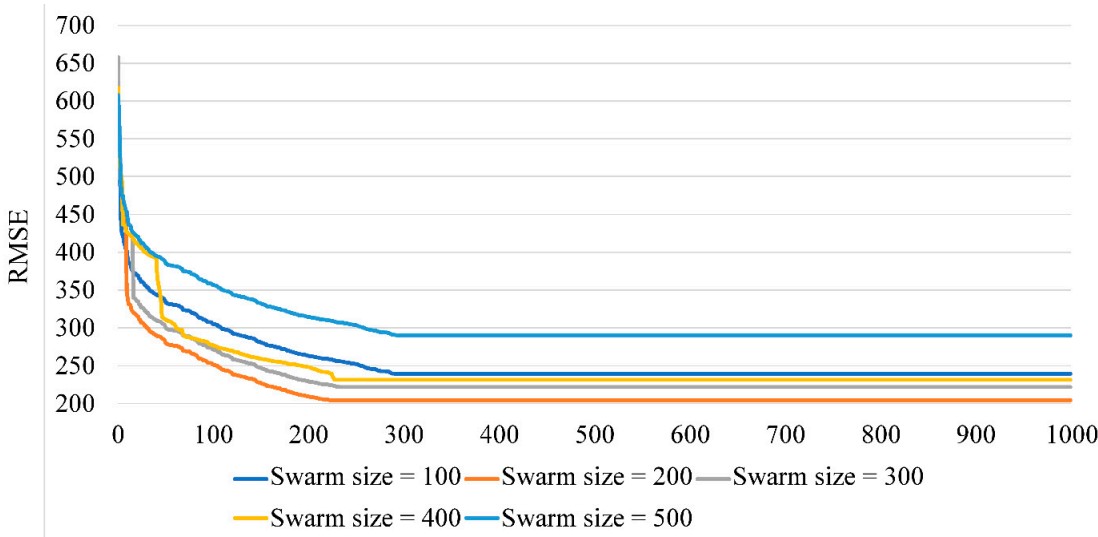

**Figure 6.** Performance of the PSO-SVR ensemble with the polynomial kernel function on the training process.

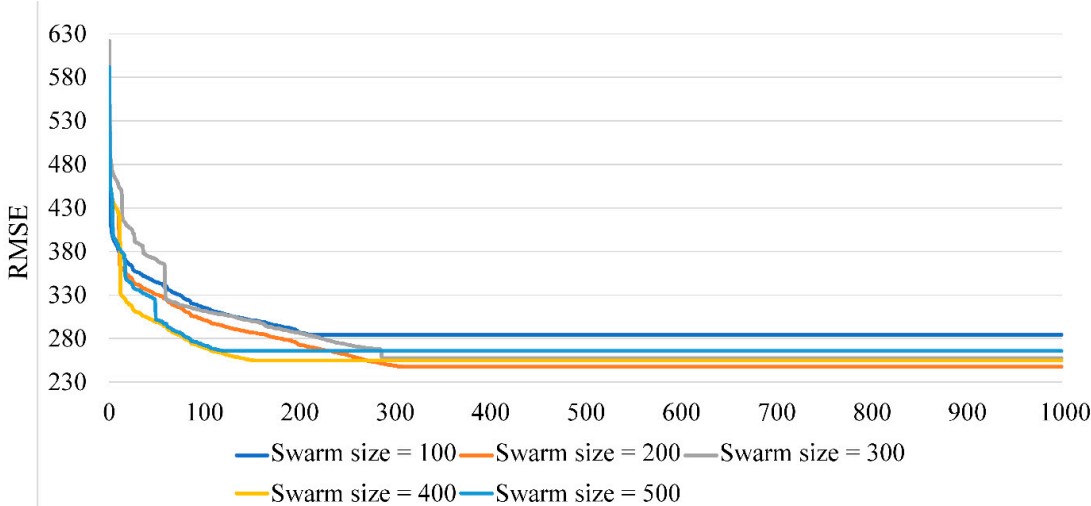

**Figure 7.** Performance of the PSO-SVR ensemble with the linear kernel function on the training process.

### 5.2. MLR Model

For MLR modeling, a simple multivariate regression technique in the Eview environment has been implemented. As a result, the MLR formula for predicting *GCV* based on the training dataset was determined as the following function:

$$GCV = -14.212M - 92.162A - 21.985VM + 8647.379 \tag{7}$$

### 5.3. CART Model

Regarding the CART model for predicting GCV, the complexity parameter (cp) was used as the main parameter for adjusting the model performance. k-fold cross-validation method was considered in the development of the CART model to avoid overfitting [105]. A grid search with cp lies in the range of 0 to 1 was established to find the optimal value of cp in the present study. Eventually, the cp value of 0 was defined as the optimal value for the CART model as shown in Figure 8.

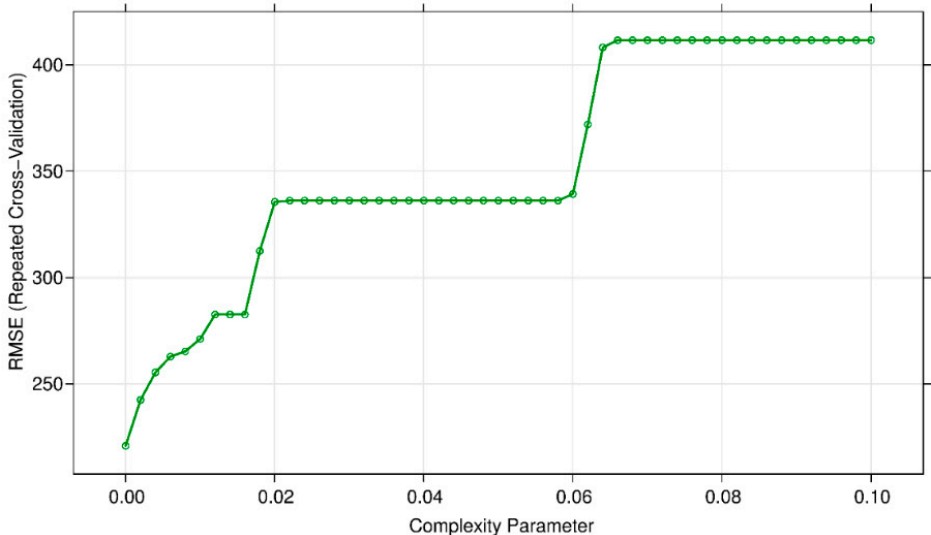

**Figure 8.** The accuracy level of the CART model with various cp.

*5.4. PCA Model*

For PCA modeling, the number of components ($\vartheta$) was used as the significant parameter for turning the performance of the PCA model. A trial and error procedure with three PCA models ($\vartheta = 1, 2$, respectively) was conducted. As a result, the best PCA model reached $\vartheta = 2$ as computed in Table 4. Note that, the k-fold cross-validation method was also used for the development of the PCA model.

**Table 4.** Performance of the PCA model with various $\vartheta$.

| $\vartheta$ | RMSE | $R^2$ |
|---|---|---|
| 1 | 475.018 | 0.755 |
| 2 | 432.723 | 0.797 |

*5.5. Evaluation*

Once the models were established based on the training dataset, their performance needed to re-checked by experimental results, i.e., the testing dataset. For this state, the testing dataset was tested by the developed models. To have a general assessment of the models, a simple ranking and intensity color methods were applied as shown in Table 5 based on the RMSE and $R^2$ values. Both training and testing processes were evaluated in this section.

**Table 5.** The results of the developed GCV predictive models.

| Model | Training | | | | Testing | | | | Total Rank |
|---|---|---|---|---|---|---|---|---|---|
| | RMSE | $R^2$ | Rank for RMSE | Rank for $R^2$ | RMSE | $R^2$ | Rank for RMSE | Rank for $R^2$ | |
| PSO-SVR-RBF | 196.878 | 0.956 | 6 | 6 | 212.831 | 0.952 | 6 | 6 | 24 |
| PSO-SVR-P | 204.347 | 0.953 | 5 | 5 | 215.767 | 0.950 | 5 | 5 | 20 |
| PSO-SVR-L | 247.581 | 0.933 | 2 | 2 | 254.216 | 0.931 | 2 | 2 | 8 |
| Multiple linear regression (MLR) | 224.283 | 0.945 | 3 | 3 | 225.943 | 0.946 | 4 | 3 | 13 |
| Classification and regression trees (CART) | 220.891 | 0.946 | 4 | 4 | 226.434 | 0.946 | 3 | 3 | 14 |
| Principal component analysis (PCA) | 432.728 | 0.797 | 1 | 1 | 439.585 | 0.794 | 1 | 1 | 4 |

The results of Table 5 revealed that the AI models provided high reliability in predicting GCV in this study. Color intensity showed that the PSO-SVR model with the radial basis kernel function (i.e., PSO-SVR-RBF) reached the highest accuracy on both training and testing datasets. Additionally, the total ranking of 24 was also interpreted the PSO-SVR-RBF performance. With a slightly lighter color intensity and the overall ranking of 20, the PSO-SVR model with polynomial kernel function (i.e., PSO-SVR-P model) yielded somewhat weaker performance than those of the PSO-SVR-RBF model. Among the PSO-SVR models with different kernel functions (i.e., RBF, P, L), the PSO-SVR model with the linear kernel function (i.e., PSO-SVR-L model) provided lowest performance with a total ranking of eight. Its performance was even lower than those of the MLR model (with an overall ranking of 13). The CART and PCA are the models that provided the poor performance in this study, especially is PCA based on the RMSE, $R^2$ and ranking values in Table 5. Indeed, the PCA model provided the weakest performance with an RMSE of 439.585, $R^2$ of 0.794, and a total ranking of four. Figures 9–14 show the deviation between the analyzed and predicted values of GCV in the developed models.

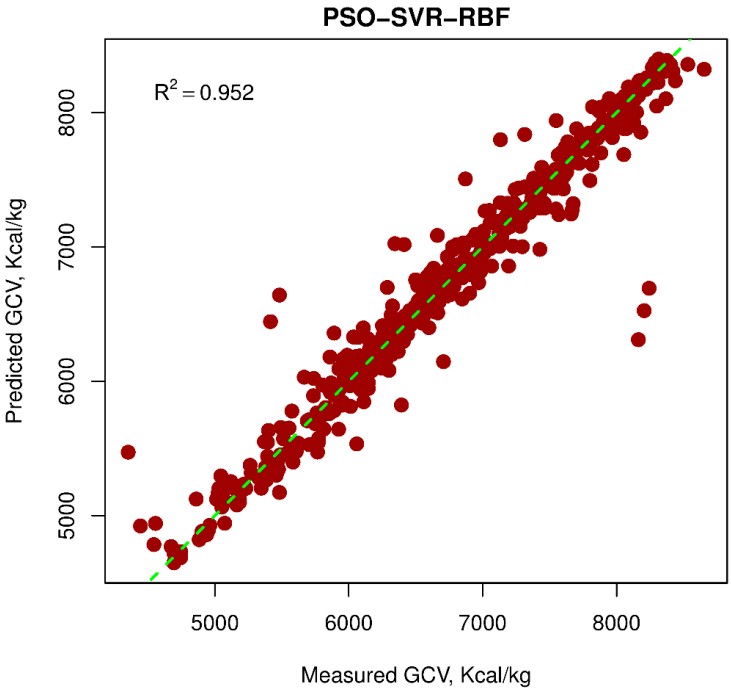

**Figure 9.** Analyzed versus predicted GCV by the PSO-SVR-RBF model.

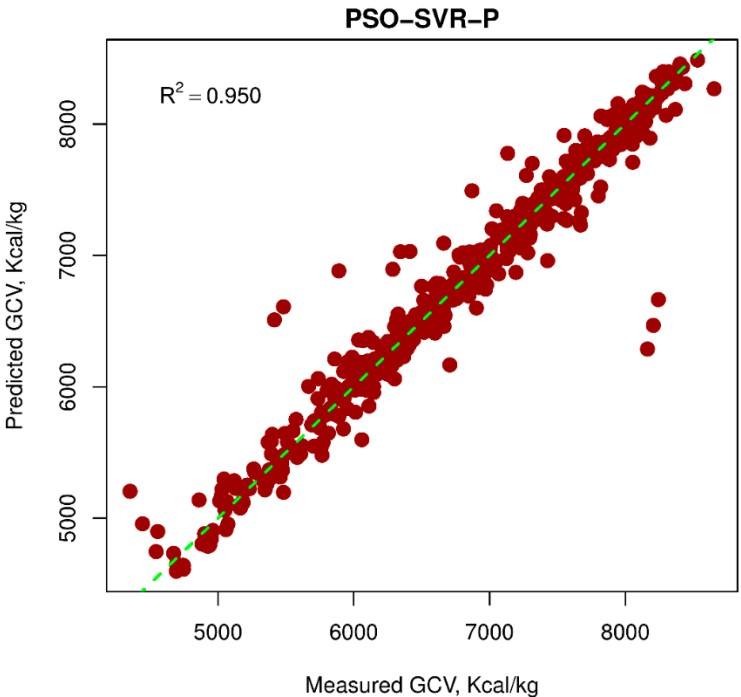

**Figure 10.** Analyzed versus predicted GCV by the PSO-SVR-P model.

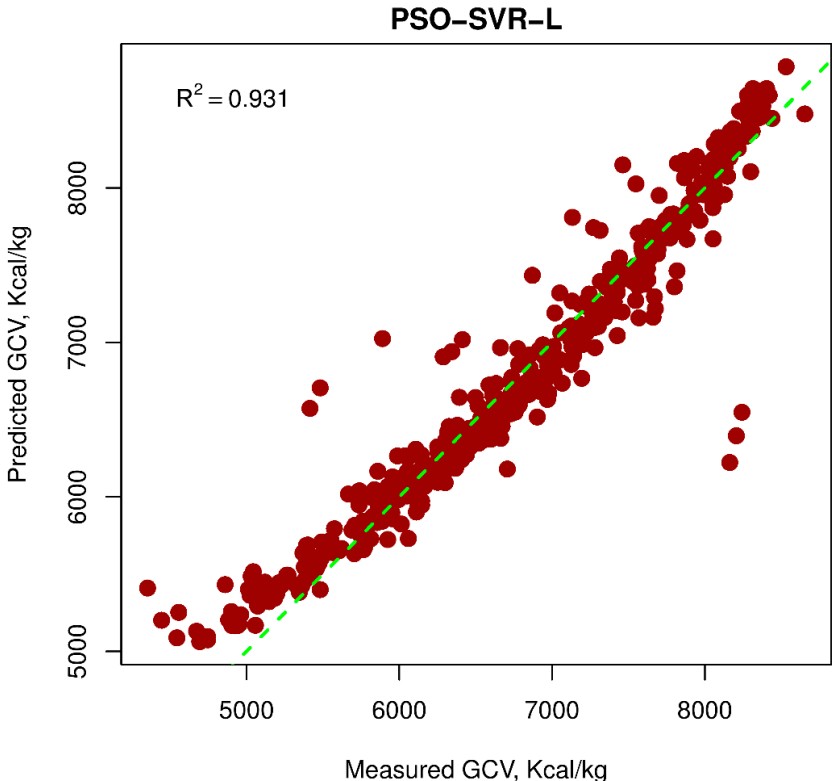

**Figure 11.** Analyzed versus predicted GCV by the PSO-SVR-L model.

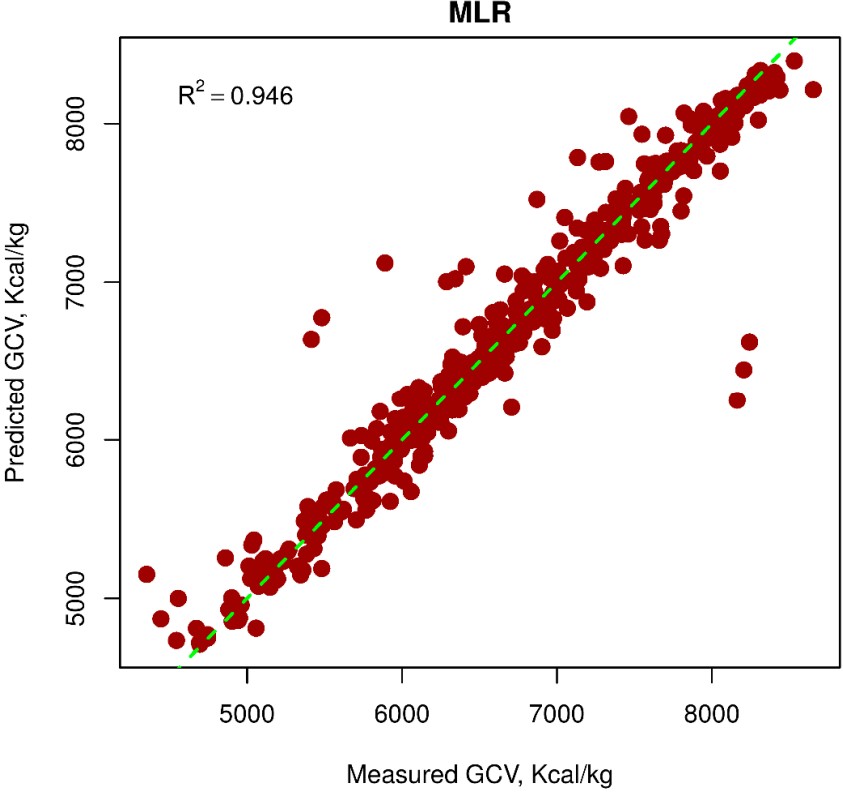

**Figure 12.** Analyzed versus predicted GCV by the MLR model.

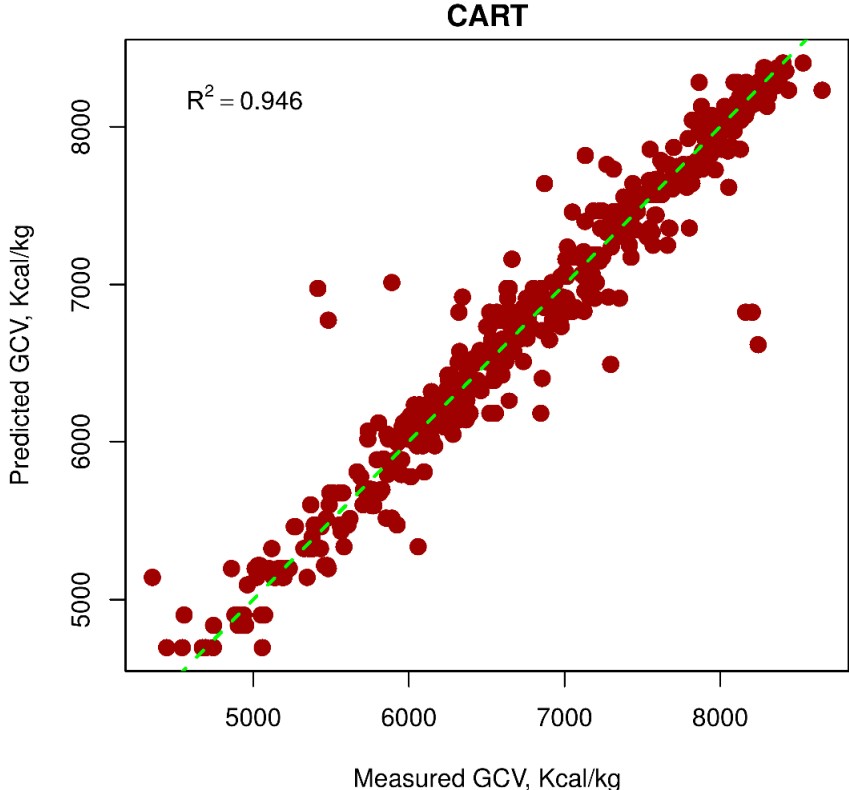

**Figure 13.** Analyzed versus predicted GCV by the CART model.

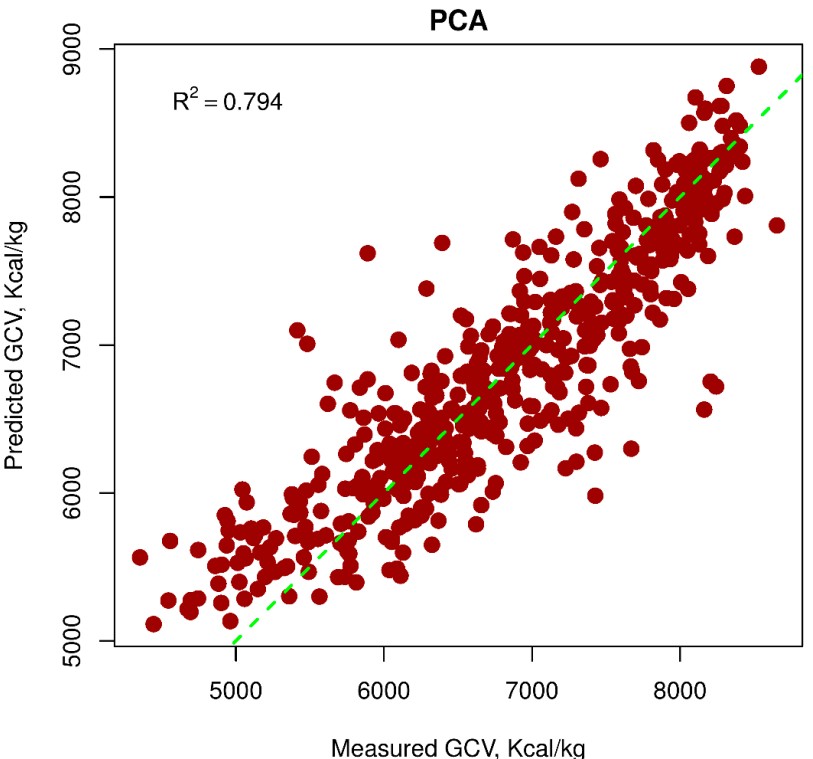

**Figure 14.** Analyzed versus predicted GCV by the PCA model.

## 6. Conclusions

Coal is one of the essential fuels for the development of countries, especially developing countries. The demand for the heat supplied by coal is enormous and needs to be calculated and forecasted. Based on the experimental analysis of coal GCV in the laboratory, this study was proposed a new hybrid AI model for predicting GCV of coal with high reliability as well as the accuracy, i.e., the PSO-SVR-RBF model. It can be considered as a robust supporting tool in estimating GCV before exploitation/using coal for heat purposes, such as thermal power plants, the metallurgy industry, and heating load systems, to name a few. The PSO algorithm implemented optimization of the SVR model by searching for the most proper hyper-parameters. Diverse kernel functions include radial, polynomial, and linear functions were tested and evaluated when combined with the SVR model and optimized by the PSO algorithm. The results showed that the RBF is the best-suited kernel function for PSO-SVR in predicting the GCV of coal in this study. This study contributed an acknowledgement as well as a new AI model (i.e., PSO-SVR-P) in statistical communities for predicting GCV with high accuracy. In addition, the developed models for estimating GCV can be considered to similarly apply to other solid fuels such as biomass or peat. However, they need further research, especially in regards to influential parameters for solid fuels. Future work can also be introduced and proposed based on the obtained results of this study. For example, the efficient energy of solid fuels, as well as renewable energy, can be predicted using potential AI models.

**Author Contributions:** Data collection and experimental works: H.-B.B., H.N., X.-N.B. Writing, discussion, analysis, and revision: H.-B.B., H.N., X.-N.B., T.N.-T., Y.C. and Y.Z.

**Funding:** This work was supported by Basic Science Research Program through the National Research Foundation of Korea (NRF) funded by the Ministry of Education (2018R1D1A1A09083947).

**Acknowledgments:** The authors would like to thank Hanoi University of Mining and Geology (HUMG), Hanoi, Vietnam; Duy Tan University, Da Nang, Vietnam; the Center for Excellence in Analysis and Experiment and the Center for Mining, Electro-Mechanical research of HUMG, Duy Tan University, Da Nang, Vietnam, and Ton Duc Thang University, Ho Chi Minh City, Vietnam.

**Conflicts of Interest:** The authors declare no conflicts of interest.

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
