# Peer review of "A Novel Artificial Intelligence Technique to Estimate the Gross Calorific Value of Coal Based on Meta-Heuristic and Support Vector Regression Algorithms"

_applsci, doi:10.3390/app9224868_

Round 1

Reviewer 1 Report

Dear Authors,

The paper presents a relevant problem of estimating GCV of coal based on meta-heuristic and SVR. 

Comments:

Figures miss legends, e.g. Fig. 2: legend is missing to describe M, A, VM...

Equations' parameters need to be described, e.g. for (1), (2), (3) e, f, c, n... need to be described.

Section 5.5 and Conclusions 6: I am missing a discussion with a comparison of your results with results gathered by other studies.

Minor mistakes:

l. 91: "stone" is typed twice

l. 147-149: ?

l. 209: "V_max,  ,, and w" - most probably, there is a typo

Author Response

Dear reviewer,

Thank you very much for your review and valuable comments/suggestions. We have revised/addressed/corrected all your comments properly. Please enclosed find the attached file.

Thanks again!

Best regards,

Hoang Nguyen

Reviewer 2 Report

The manuscript deals with an evolutionary-based predictive system for determining gross calorific value. Even if many papers were already published during the last years on the following topic, the present work proposes an original approach. In general, it is to some extent interesting and easy to follow. However, in my opinion, the novelty and scientific soundness of the paper should be described better in the introduction section.

On page 14, authors show that the PSO-SVR-P model reached the highest accuracy. In case of that model, would it not be beneficial to describe what are the differences between measured and estimated GCV for the less accurate cases? Could any of developed models be adopted for estimation of GCV of other solid fuels such as biomass or peat?

I do not feel qualified to judge about the English language and style, nevertheless the text needs some revision, for e.g. there are missing units in table 1, in line 46 there is “nitro” instead of nitrogen, in line 91 the word “sandstone” is unnecessarily repeated, in line 461 capital letters are not necessary for the names of authors, etc.

Author Response

(The authors gave the same response as above.)

Round 2

Reviewer 1 Report

Dear Authors,

thank you for considering the comments and suggestions.